# c-di-GMP heterogeneity is generated by the chemotaxis machinery to regulate flagellar motility

**Bridget R Kulasekara[1], Cassandra Kamischke[2], Hemantha D Kulasekara[2], Matthias Christen[2†], Paul A Wiggins[3,4], Samuel I Miller[2,5,6]\***

[1]Molecular and Cellular Biology Program, University of Washington, Seattle, United States; [2]Department of Microbiology, University of Washington, Seattle, United States; [3]Department of Physics, University of Washington, Seattle, United States; [4]Department of Bioengineering, University of Washington, Seattle, United States; [5]Department of Medicine, University of Washington, Seattle, United States; [6]Department of Genome Sciences, University of Washington, Seattle, United States

**Abstract** Individual cell heterogeneity is commonly observed within populations, although its molecular basis is largely unknown. Previously, using FRET-based microscopy, we observed heterogeneity in cellular c-di-GMP levels. In this study, we show that c-di-GMP heterogeneity in *Pseudomonas aeruginosa* is promoted by a specific phosphodiesterase partitioned after cell division. We found that subcellular localization and reduction of c-di-GMP levels by this phosphodiesterase is dependent on the histidine kinase component of the chemotaxis machinery, CheA, and its phosphorylation state. Therefore, individual cell heterogeneity in c-di-GMP concentrations is regulated by the activity and the asymmetrical inheritance of the chemotaxis organelle after cell division. c-di-GMP heterogeneity results in a diversity of motility behaviors. The generation of diverse intracellular concentrations of c-di-GMP by asymmetric partitioning is likely important to the success and survival of bacterial populations within the environment by allowing a variety of motility behaviors.

**\*For correspondence:** millersi@uw.edu

**†Present address:** Institute of Molecular Systems Biology, Department of Biology, ETH Zurich, Zurich, Switzerland

**Competing interests:** The authors declare that no competing interests exist.

## Introduction

Single cells in an isogenic population display heterogeneity in a variety of physiological parameters including growth rate, chemotaxis, metabolism, nutritional acquisition, and tolerance to noxious stimuli including antibiotics (*Balaban et al., 2004*; *Shibata and Ueda, 2008*; *Lidstrom and Konopka, 2010*; *Wakamoto et al., 2013*). Second messenger-based signaling, having a global impact on cellular physiology (*Romling et al., 2013*), can be a mechanism by which environmental signals are rapidly translated into phenotypic heterogeneity. However, such mechanisms for generating phenotypic heterogeneity have yet to be described for many cell types, including bacteria. Nucleotide-based second messengers including cAMP and cyclic dinucleotides perform crucial functions within prokaryotes (*Corrigan and Grundling, 2013*; *Kalia et al., 2013*). The bacterial second messenger c-di-GMP is synthesized and degraded by diguanylate cyclases (DGCs) and phosphodiesterases (PDEs) to regulate diverse processes including cell-cycle progression, motility, and exopolysaccharide production (*Romling et al., 2013*). Traditional bulk culture-based biochemical measurements cannot determine the variation of second messenger levels within populations. To measure c-di-GMP concentrations in individual cells, our laboratory developed a genetically encoded FRET-based biosensor using the *Salmonella* Typhimurium c-di-GMP binding protein YcgR (*Christen et al., 2010*). FRET microscopy analysis using this biosensor revealed that the concentration of the second messenger c-di-GMP varies bimodally in populations of diverse bacterial species (*Christen et al., 2010*).

**eLife digest** Bacterial populations have traditionally been assumed to be made up of identical cells. However, while the bacteria within a population may be genetically identical, individual cells have different growth rates, metabolisms and motilities, among other things. This 'phenotypic heterogeneity' has been observed in many different species of bacteria, and in some cases it can be attributed to changes in the concentration of molecules called second messengers that help to relay signals from the external environment to targets within the cell.

It can be challenging to monitor changes in the concentration of specific molecules inside cells, but researchers recently developed a form of microscopy based on FRET (short for Forster resonance energy transfer) that can measure the levels of a second messenger molecule called cyclic di-guanylate (c-di-GMP) inside individual cells. This technique was used to study *P. aeruginosa*, a bacterium that has a single corkscrew-shaped propeller that enables it to swim through liquid. *P. aeruginosa* divides to form two daughter cells—one with a propeller and one without.

Although the daughter cell that does not have a propeller quickly grows one, FRET-based microscopy revealed that the daughter cell with a propeller had less c-di-GMP than the daughter without a propeller, but the reasons underlying this difference and its effects on bacterial behavior were not clear.

Now Kulasekara et al. show that the cell that inherits the propeller contains an enzyme that degrades c-di-GMP, and that the low levels of this second messenger molecule—caused by the enzyme being concentrated near the base of the propeller, and the presence of a protein (CheA) that enables the bacteria to swim towards sources of nutrients—result in faster swimming speeds and increased responsiveness to nutrients. In other words, although the two daughter cells are genetically identical, they behave quite differently because of the different levels of this second messenger molecule.

The existence of heterogeneity within a bacterial population likely leads to increased success and survival within changing diverse environments, and this work sets the stage for similar investigations into what establishes heterogeneity in other bacterial populations.

---

One species we examined, *Caulobacter crescentus*, in contrast to other well-studied gram-negative bacteria, exhibits an asymmetric cell cycle in that the two daughter cells have unique morphologies and functions. Only the daughter swarmer cell with a polar flagellum is motile, and only the stalk cell, with the exopolysaccharide containing holdfast, can undergo cell division (*Tsokos and Laub, 2012*). It has been known that a c-di-GMP synthesizing enzyme, PleD, is localized and activated in the stalked cell (*Paul et al., 2004*; *Aldridge et al., 2003*). Therefore, bimodal distribution of c-di-GMP was not a surprising observation for *C. crescentus*. However, the features that facilitate preferential PleD localization to the stalked cell pole are undetermined. Other species we examined, including *Pseudomonas aeruginosa*, *Salmonella* Typhimurium, and *Klebsiella pneumonia* (*Christen et al., 2010*), all produce morphologically similar progeny. Therefore, heterogeneity in c-di-GMP levels for these bacteria was a surprising observation.

Other than the example of *C. crescentus*, little is known about individual variation of microbes as a product of cell division. Often the possibility of heterogeneity is evoked to explain surprising phenotypes, such as variability in the susceptibility of bacterial populations to antibiotics (*Dhar and McKinney, 2007*). Current knowledge suggests variation resulting from cell division could occur through several mechanisms including inherent cellular polarity (*Dworkin, 2009*). Cellular heterogeneity can also result from processes that are stochastic including mutations, unequal partitioning of less abundant proteins (*Elowitz et al., 2002*), or alterations in gene expression as a result of phase variation (*Henderson et al., 1999*).

In *P. aeruginosa*, although cellular differentiation is not morphologically obvious, cells possess a single polar flagellum. The daughter cell that does not inherit the flagellum following cell division rapidly synthesizes a new flagellum (*Suzuki and Iino, 1980*). The daughter cell with the lower c-di-GMP concentration after cell division was previously demonstrated to inherit the single polar flagellum, suggesting this asymmetry is not a stochastic process (*Christen et al., 2010*). In this work, we define the mechanism that generates *P. aeruginosa* c-di-GMP heterogeneity during the cell cycle. We have discovered that

the chemotaxis machinery, the signal transducing system required for directed bacterial navigation, additionally activates a PDE to generate low c-di-GMP levels in flagellated cells. This PDE is localized to the flagellated cell pole by the chemotaxis machinery, indicating that asymmetry in organelle distribution during cell division results in the bimodal distribution of c-di-GMP. This heterogeneity in c-di-GMP levels in turn controls flagellar-based motility behavior.

## Results

### A specific PDE, encoded by PA5017, is required for c-di-GMP heterogeneity

Using FRET-based microscopy, we imaged cells from single time points to visualize heterogeneity in c-di-GMP levels. Use of an automated method to segment cells and analyze their fluorescence intensity allowed us to determine that 20% of wild type *P. aeruginosa* PA14 cells reproducibly exhibit c-di-GMP concentrations less than 200 nM during exponential growth (*Figure 1A,B,G*). *P. aeruginosa* encodes multiple isoforms of DGCs and PDEs. We hypothesized that the activity of one or more of these c-di-GMP metabolizing enzymes is responsible for c-di-GMP heterogeneity and therefore screened a transposon mutant bank for mutants with altered levels of c-di-GMP. Out of a potential 38 enzyme homologs, we did not identify any single *dgc* genes required to maintain the population of cells with high c-di-GMP, suggesting more than one DGC is involved in synthesizing c-di-GMP during exponential growth. However, we identified one gene, PA5017, responsible for maintaining the population of cells with reduced c-di-GMP levels. Previous characterization has shown that the protein encoded by this gene is a PDE (*Roy et al., 2012*). In-frame deletion of PA5017 (*Figure 1C,D,G*) and gene complementation experiments (*Figure 1E–G*), including those using a catalytically inactive mutant, confirmed that the PDE activity of PA5017 is required for a c-di-GMP bimodal distribution. We subsequently refer to PA5017 as *pch,* as it is a phosphodiesterase determinant of c-di-GMP heterogeneity.

### Polar localization of the Pch PDE during cell division generates cells with diverse c-di-GMP levels

We hypothesized that c-di-GMP heterogeneity could occur by the asymmetric partitioning of Pch following cell division. Therefore, we constructed a functional Pch-mCherry fusion to simultaneously characterize the location of Pch and measure cellular c-di-GMP levels. Pch-mCherry exhibited polar localization in 50% of cells (*Figure 2A*). Using time-lapse FRET imaging, we observed that polarly localized Pch-mCherry is partitioned to a single cell following cell division (*Figure 2B*). Furthermore, Pch-mCherry localizes to the pole of the incipient daughter cell that exhibits reduced cellular c-di-GMP following cell division (*Figure 2B*). We quantified the association between Pch-mCherry localization and c-di-GMP levels from single time point FRET microscopy images (*Figure 2—figure supplement 1*). Cells were separated into two groups according to c-di-GMP levels and the polar intensity of Pch-mCherry was plotted for each group. Few cells were found to demonstrate high levels of c-di-GMP and to have Pch-mCherry at the cell pole. Overall, cells with c-di-GMP levels less than 200 nM exhibited high amounts of Pch-mCherry at the pole (the median polar intensity was increased by 63 ± 23%) (*Figure 2C*). These results indicate that c-di-GMP heterogeneity occurs mostly through differential polar localization and the subsequent asymmetric partitioning of Pch during cell division.

### The Pch PDE requires the CheA component of the chemotaxis machinery to localize to the pole

During cell division, only one progeny cell inherits the already formed single polar flagellum (*Suzuki and Iino, 1980*). *P. aeruginosa* cells containing low c-di-GMP following cell division also inherit the single polar flagellum (*Christen et al., 2010*), suggesting that c-di-GMP heterogeneity in *P. aeruginosa* may require the flagellum. Therefore, we examined Pch-mCherry localization in mutants of the flagellar rotor components, FliM, FliN, FliG, and a flagellar secretory component, FlhB (*Dasgupta et al., 2003*). Pch-mCherry localization and foci formation was abolished in these mutants (*Figure 3—figure supplement 1*). We quantified these observations using a strain lacking FliF, an inner membrane component of the MS-ring. The *fliF* mutant has no flagella (*Arora et al., 1996*; *Li and Sourjik, 2011*). Localization, or foci formation, of Pch-mCherry was abolished in the *fliF* mutant (*Figure 3A,C*) and biosensor activity measurements revealed that more than 99% of Δ*fliF* cells exhibit c-di-GMP levels greater than 200 nM (*Figure 3B*). Measurements of mean cellular Pch-mCherry intensity in this strain

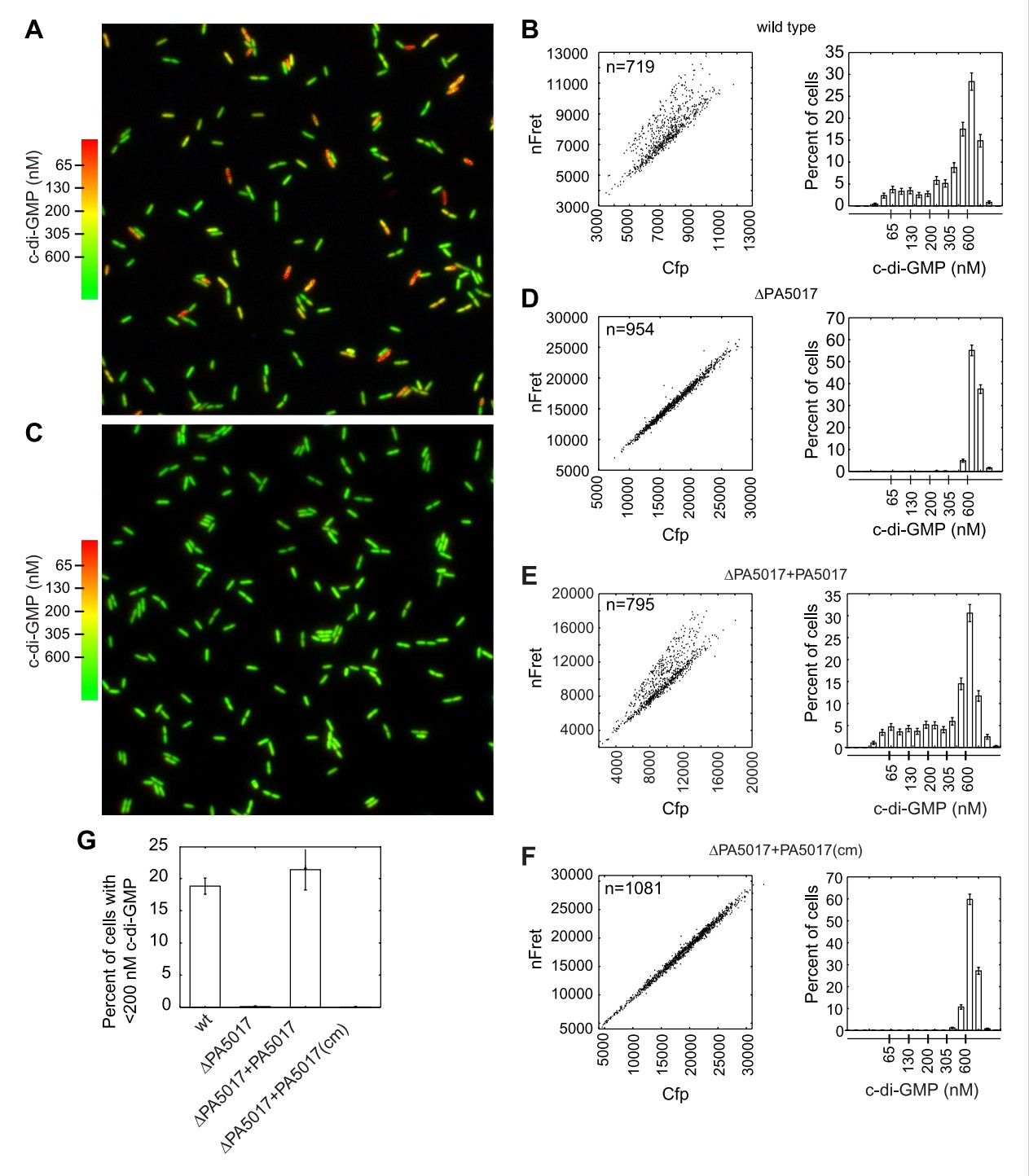

**Figure 1**. PA5017/*pch* is required for heterogeneity in c-di-GMP levels. (**A**) Wild-type *P. aeruginosa* cells exhibit heterogeneity in c-di-GMP levels. Pseudocolored nFret/Cfp ratios from a single field (88.7 × 88.7 microns) demonstrate c-di-GMP concentrations in wild-type *P. aeruginosa* PA14 cells. (**B**) Quantification of cellular c-di-GMP levels in wild-type *P. aeruginosa* cells. The left panel shows a scatter plot of the mean nFret vs Cfp values in individual cells. In the right panel is a histogram of the corresponding cellular c-di-GMP levels where error bars depict the counting error. Graphs in (**D**), (**E**), and (**F**) assume the same format. (**C**) c-di-GMP concentrations in the PA5017 deletion. (**D**) Quantification of c-di-GMP concentrations in ΔPA5017 cells. (**E**) Quantification of c-di-GMP concentrations in ΔPA5017 complemented with PA5017. (**F**) Quantification of c-di-GMP concentrations in ΔPA5017 complemented with a catalytic mutant (cm) of PA5017. (**G**) Quantification of the mean percentage of cells with less than 200 nM c-di-GMP from three biological replicates. Error bars depict the standard deviation.

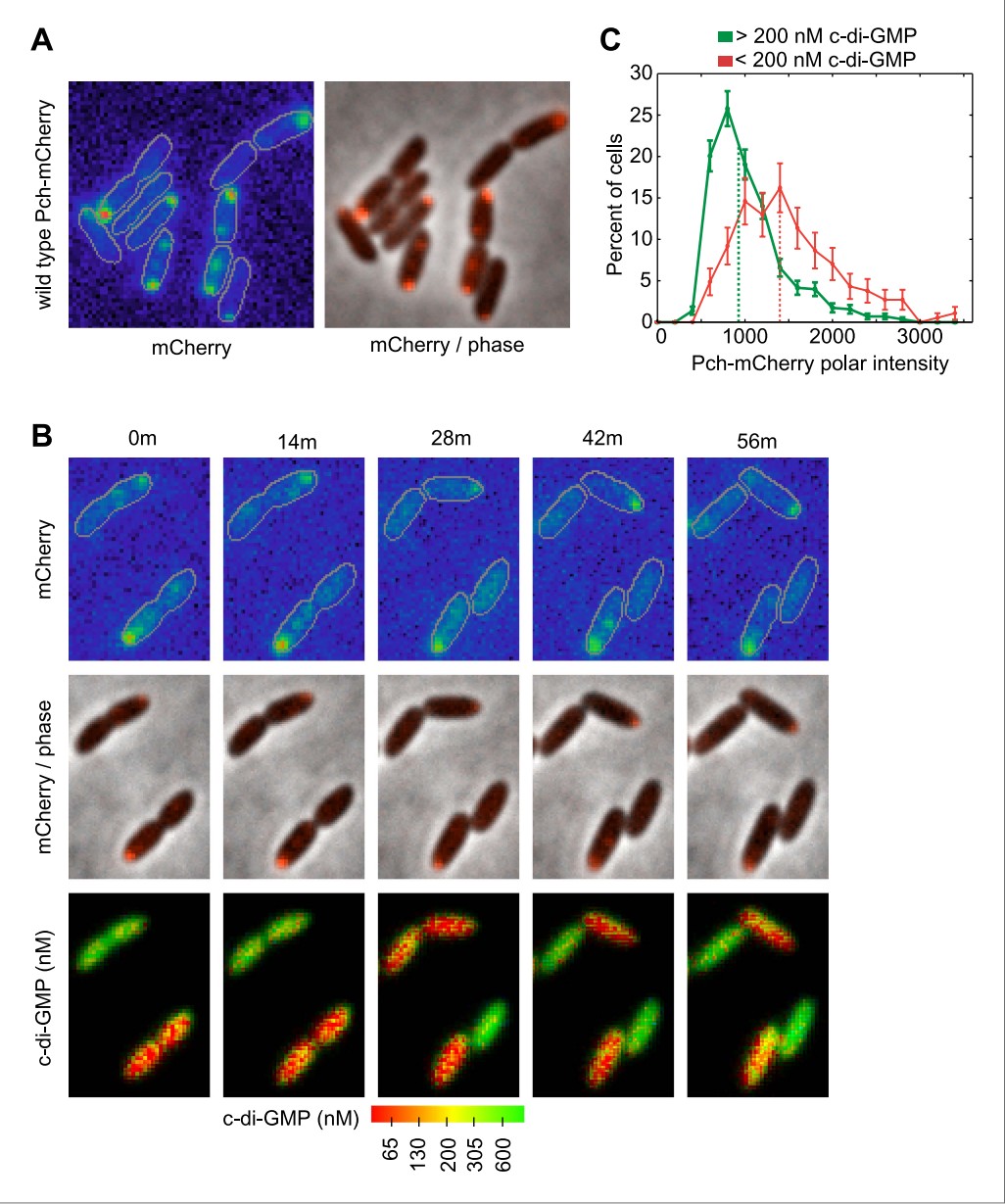

**Figure 2**. Pch localizes to the cell pole and is asymmetrically partitioned to generate heterogeneity in cellular c-di-GMP levels. (**A**) A representative image (10.8 × 10.8 microns) of Pch-mCherry subcellular localization. The fluorescence channel (mCherry) on the left illustrates the range of subcellular concentrations of Pch-mCherry. A rainbow color map was utilized to represent different intensity values. Red represents the highest intensity and purple represents the lowest intensity. Cell boundaries are delineated in gray. An overlay of the fluorescence channel (in red) and the phase contrast image on the right, illustrates the predominant Pch-mCherry subcellular localization pattern. Refer to the 'Materials and methods' section for a description of contrast settings for both image types. (**B**) Polar localization of Pch-mCherry is associated with low cellular c-di-GMP after cell division, as shown by representative time-lapse images (width of 6.3 microns) of biosensor activity and Pch-mCherry localization in two dividing cells. The top panel is of the fluorescence image. The middle panel contains an overlay of the fluorescence channel (in red) and the phase contrast image. The bottom panel displays pseudocolored nFret/Cfp values that depict c-di-GMP concentrations. (**C**). Distribution of the mean polar Pch-mCherry intensity of 797 cells binned according to c-di-GMP. Polar intensity has been plotted for the cell pole with the greatest intensity. Dotted lines mark the median values. Error bars depict the counting error.

*Figure 2. Continued on next page*

*Figure 2. Continued*

The following figure supplements are available for figure 2:

**Figure supplement 1**. Polar localization of Pch-mCherry is associated with lower cellular c-di-GMP.

background (*Figure 3—figure supplement 2*) indicated this strain has decreased levels of Pch-mCherry. It is therefore not clear whether c-di-GMP is elevated in the *fliF* mutant because of reduced levels of Pch or because Pch is inactive when delocalized.

To determine whether a polarly placed flagellum is required for Pch localization, we measured the effect of deleting *flhF*. FlhF is a GTPase required for polar positioning of the flagellum, and an *flhF* deletion causes the flagellum to delocalize from the pole in *P. aeruginosa* (*Murray et al., 2006*). We observed that an *flhF* deletion also causes Pch-mCherry foci to delocalize from the pole (*Figure 3A,C,D*). Nevertheless, biosensor activity showed that *flhF* mutant cells are able to maintain c-di-GMP heterogeneity (*Figure 3B*, right panel). Thus, the flagellum, but not its polar positioning, is required for the maintenance of low c-di-GMP concentrations. These data suggested that Pch may bind to a flagellar component. However, an intact flagellar basal body rod-hook structure is additionally required for expression of the chemotaxis machinery (*Dasgupta et al., 2003*), which is also polarly localized in *P. aeruginosa* (*Guvener et al., 2006*). We therefore systematically deleted *che* genes to determine whether the chemotaxis machinery is required for polar localization of Pch. Deletions of the methyl accepting chemoreceptor (MCP) methyltransferase, *cheR*, and chemotaxis machinery response regulators, *cheY* and *cheB*, did not have any effect on Pch-mCherry localization (*Figure 4A*). However, we discovered that a *cheA* null strain lacks Pch-mCherry localization (*Figure 4C*) and also fails to demonstrate c-di-GMP heterogeneity (*Figure 4B,D*). Moreover, mean cellular intensity of Pch-mCherry in a *cheA* mutant is similar to that of wild type and exceeds that of the *flhF* mutant (*Figure 3—figure supplement 2*). Taken together, these results indicate that the chemotaxis machinery histidine kinase, CheA, is required for both polar localization and activity of Pch.

## The phosphorylated form of the sensor kinase CheA promotes Pch activity

The chemotaxis machinery is a signal transduction system that facilitates directed navigation in the presence of chemical gradients and random navigation in uniform environments (*Wadhams and Armitage, 2004*; *Sourjik and Wingreen, 2012*). For many species of bacteria, counter clockwise rotation of the flagellum propels the cell forward in a smooth, straight trajectory, and clockwise rotation causes the cell to tumble and reorient (*Berg and Brown, 1972*) or, in the case of the cell with a unipolar flagellum, to reverse and then reorient (*Wadhams and Armitage, 2004*; *Xie et al., 2011*; *Qian et al., 2013*). The chemotaxis machinery modulates the bacterial trajectory by controlling the frequency of clockwise rotational events. Clockwise rotation is initiated by phosphorylated CheY binding to the flagellar rotor (*Baker et al., 2006*) and MCPs control the rate of CheY phosphorylation by CheA (*Vladimirov and Sourjik, 2009*; *Hazelbauer and Lai, 2010*). Constitutively active CheR methylates MCPs, of which *P. aeruginosa* possesses 26, whereas phosphorylated CheB demethylates MCPs (*Baker et al., 2006*). Methylation specifically decreases MCP affinity for its ligand (*Sourjik and Berg, 2002*) and increases CheA kinase activity. MCP methylation functions to provide memory of the past few seconds of the cell's environment and render cells sensitive to changes relative to recent conditions. When cells are in a uniform environment, the resulting steady-state kinase activity initiates a low basal rate of cell reversals, conferring a random walk trajectory (*Berg and Brown, 1972*; *Vladimirov and Sourjik, 2009*). As CheA is required to generate low c-di-GMP levels, we thought it possible that the steady-state activity of CheA also affects c-di-GMP heterogeneity.

To test whether steady-state kinase activity is required for maintaining c-di-GMP heterogeneity, we generated a strain in which the CheA histidine-49 residue required for CheA autophosphorylation and subsequent phosphotransfer to a response regulator has been mutated to an asparagine. We found that this phosphotransfer mutant conferred by the CheAH49N allele had no effect on Pch-mCherry localization (*Figure 4A*). However, c-di-GMP levels increased relative to wild type, as more than 90% of cells exhibited greater than 200 nM intracellular c-di-GMP, decreasing heterogeneity (*Figure 4B*). We obtained a similar result for the *cheR* deletion strain, which confirms the result from the CheA kinase mutant because CheR stimulates auto-kinase activity of CheA. The requirement of CheA histidine-49

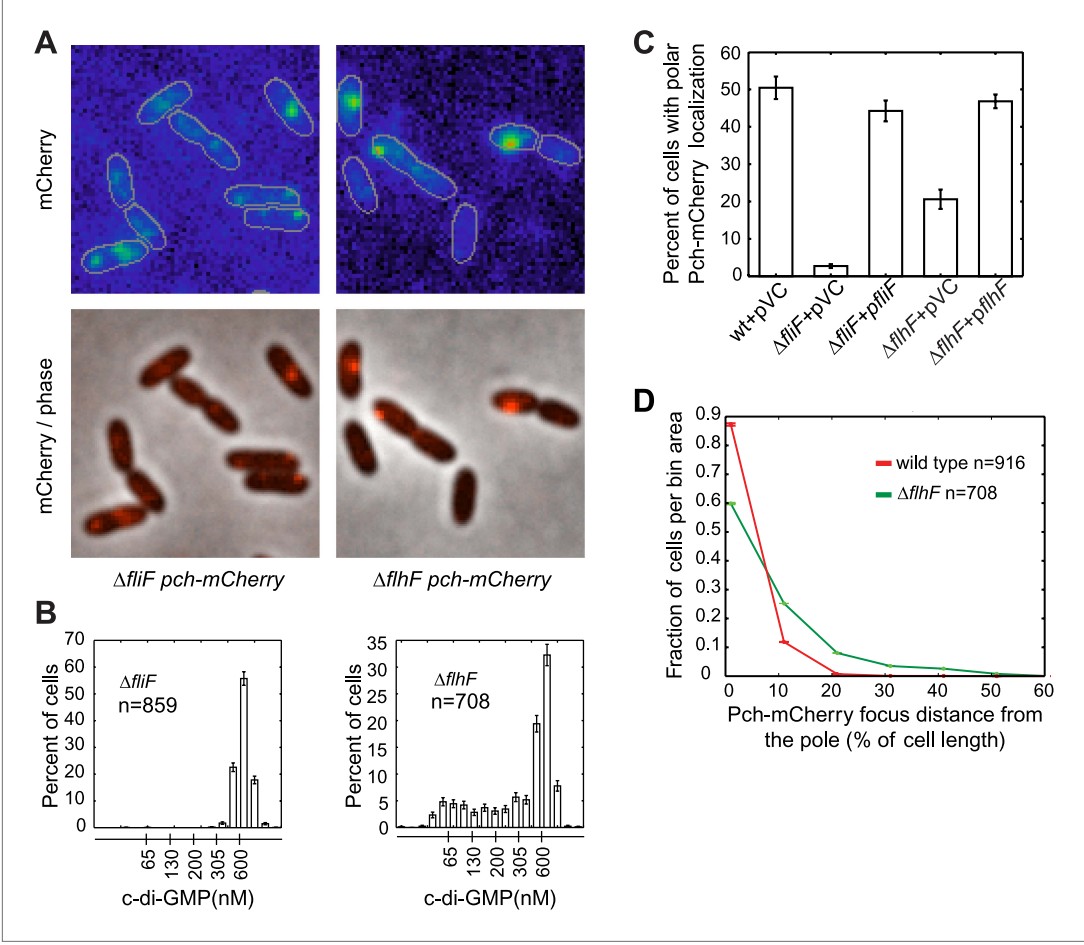

**Figure 3**. The flagellar apparatus is required for Pch polar localization and maintenance of low c-di-GMP. (**A**) Representative images (10.8 × 10.8 microns) of Pch-mCherry localization in *fliF* and *flhF* deletions. Refer to the 'Materials and methods' section for a description of contrast settings for both image types. (**B**) Histograms of cellular c-di-GMP concentrations in *fliF* and *flhF* deletions. Error bars represent counting error. The mean percentage of Δ*fliF* cells with less than 200 nM c-di-GMP from three biological replicates is 0.1% (standard deviation of 0.1%). The mean percentage of Δ*flhF* cells with less than 200 nM c-di-GMP from three biological replicates is 26.6% (standard deviation of 5.2%). (**C**) Quantification of the mean percentage of cells exhibiting polar localization of Pch-mCherry in Δ*fliF* and Δ*flhF* backgrounds from four biological replicates. Error bars represent the standard deviation. Strains contain empty vector (pVC) or a complementing plasmid. Quantification was performed as described in the 'Materials and methods'. (**D**) A representative histogram of Pch-mCherry focus distance to the nearest cell pole in wild type and *flhF* deletion strains. Error bars depict counting error.

The following figure supplements are available for figure 3:

**Figure supplement 1**. The flagellar apparatus is required for Pch-mCherry polar localization.

**Figure supplement 2**. Quantitation of Pch.

for the maintenance of low c-di-GMP indicates that the chemotaxis apparatus controls c-di-GMP heterogeneity in a manner dependent on the phosphorylation of CheA.

## The Pch PDE forms a complex with CheA at the flagellated pole

As the above data suggested Pch forms a complex with CheA, we subsequently performed co-localization studies of Pch and CheA. To determine whether Pch co-localizes with CheA, we constructed and imaged functional CheA-mTurquoise (CheA-mTq) and Pch-Yfp fusions. The majority (93 ± 1%) of cells containing both CheA and Pch fusions had these proteins co-localized to the same pole (***Figure 5A,B***).

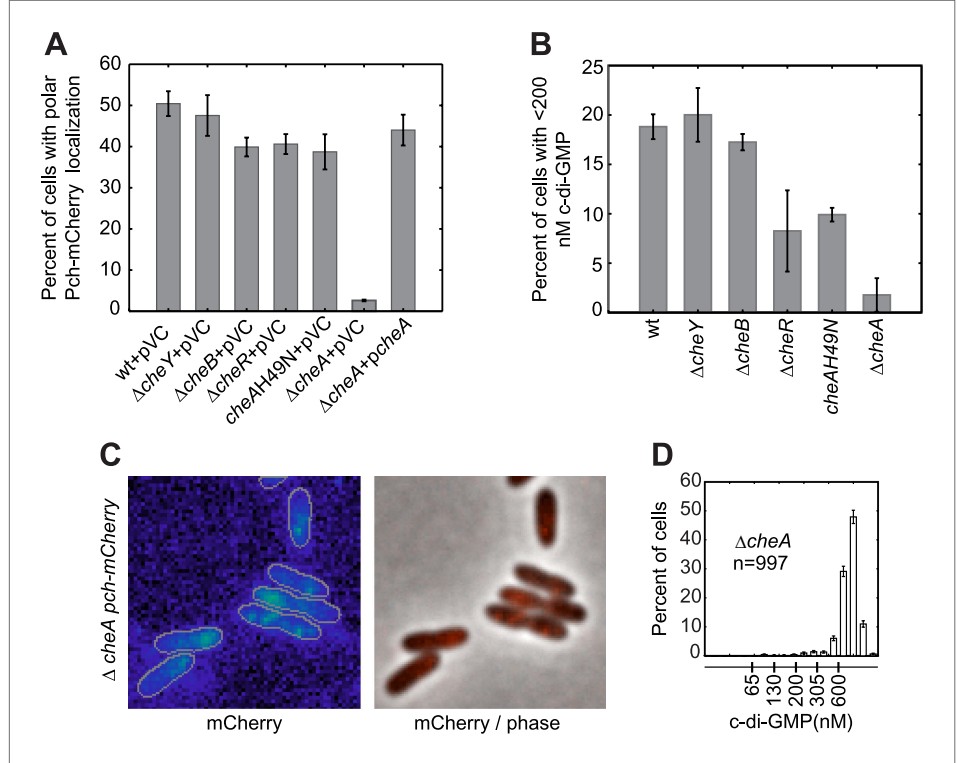

**Figure 4**. CheA is required for the maintenance of low c-di-GMP and polar localization of Pch-mCherry. (**A**) Quantification of Pch-mCherry polar localization in different *che* mutants. Strains contain empty vector (pVC) or a complementing plasmid. (**B**) Quantification of c-di-GMP in different *che* mutants. Data plotted in (**A**) and (**B**) are from three biological replicates. Error bars in (**A**) and (**B**) depict standard deviation. (**C**) Representative images (10.8 × 10.8 microns) of Pch-mCherry localization in a *cheA* deletion. (**D**) A histogram of cellular c-di-GMP levels in Δ*cheA* cells. Error bars depict the counting error.

Furthermore, the *flhF* deletion caused Pch-Yfp and CheA-mTq to delocalize from the pole but to still associate (*Figure 5A–D*). The identity of the pole the chemotaxis machinery assembles at was unknown; therefore, we simultaneously measured localization of CheA and FliM, a flagellar rotor protein. A functional CheA-mTq fusion localized to the same pole as a functional FliM-mKate2 fusion in 87 ± 3% of cells (*Figure 5—figure supplement 1A–C*). However, these two protein complexes were separated by a measurable distance as indicated by a peak in the histogram (median distance of 0.190 ± 0.010 μm) in contrast to the undetectable spacing between Pch-Yfp or Pch-mCherry and CheA-mTq (*Figure 5B*, *Figure 5—figure supplement 2*). These co-localization studies indicate Pch is associated with the chemotaxis machinery but not with the flagellar apparatus present at the same pole. As previous observations have shown that a *cheA* mutant is still motile (*Ferrandez et al., 2002*), and a *cheA* deletion does not affect FliM-mKate2 polar localization (*Figure 5—figure supplement 3*), these observations indicate that a *cheA* deletion does not affect flagellar assembly and that that Pch activation and localization is mostly dependent on CheA.

The co-localization analysis suggested that CheA and Pch may form a complex. We subsequently attempted co-immunoprecipitation (Co-IP) of CheA and Pch from a strain with a chromosomally encoded CheA-mTq, a plasmid encoded Pch fused with the VSV-G epitope tag, and a *pch* deletion. We found CheA-mTq specifically precipitates with Pch-VSV-G (*Figure 5E*). Taken together, these results support the conclusion that Pch binds to the chemotaxis machinery protein CheA or to another adaptor protein in complex with CheA.

## c-di-GMP reduces *P. aeruginosa* flagellar velocity and reversals

In multiple species, bulk, population-based assays have been used to characterize the effect of c-di-GMP on motility (*Girgis et al., 2007*; *Liu et al., 2010*; *Chen et al., 2012*). In many instances, the precise

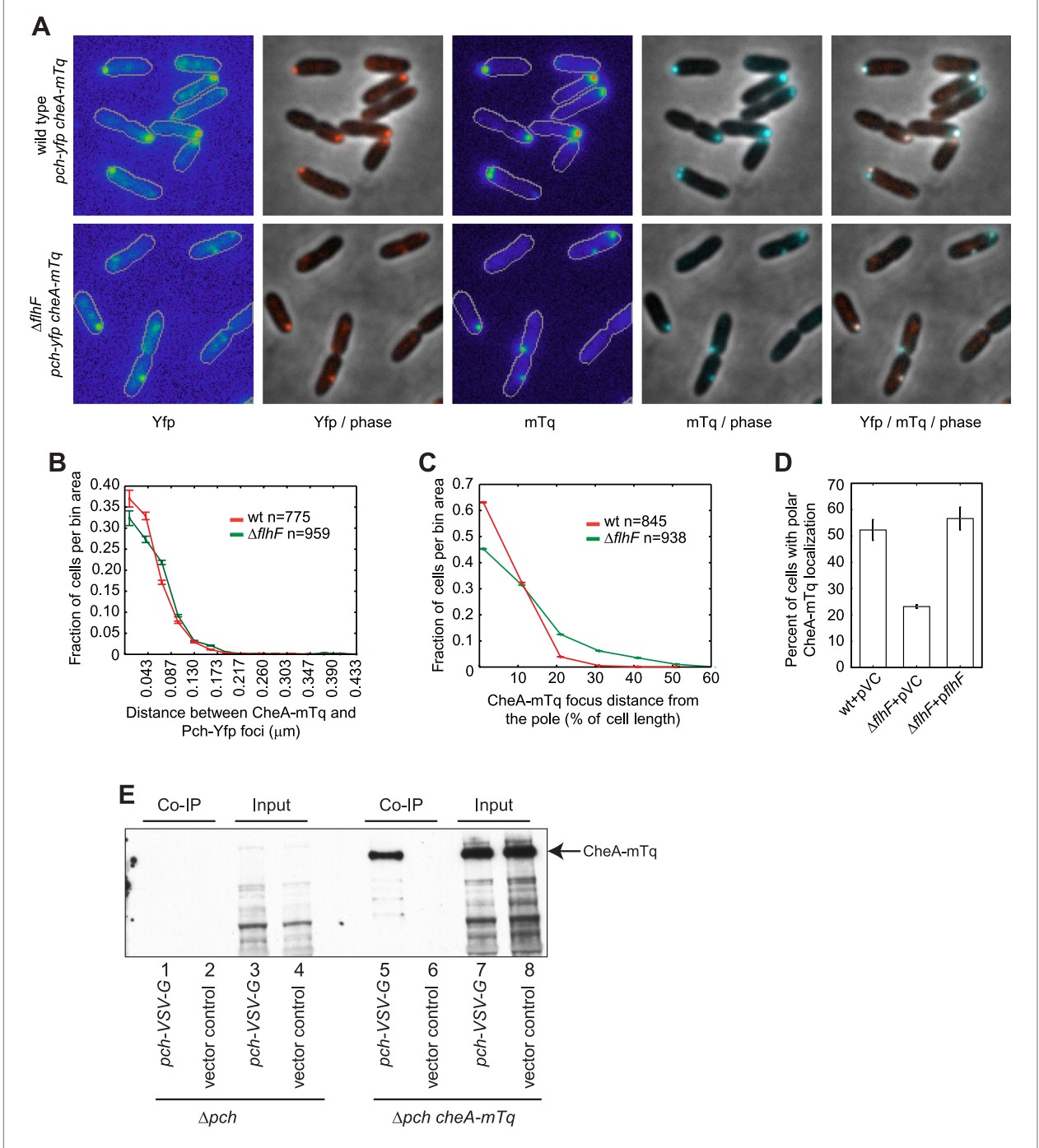

**Figure 5.** CheA forms a complex with Pch and relies on *flhF* for polar placement. (**A**) Representative images (10.8 × 10.8 microns) of CheA-mTq and Pch-Yfp localization in wild-type (top) and Δ*flhF* (bottom) strains. In the phase/fluorescence channel overlays, CheA-mTq fluorescence is shown in blue and Pch-Yfp fluorescence is shown in red. (**B**) Histogram of the smallest distance between any two CheA-mTq and Pch-Yfp foci in wild-type and Δ*flhF* strain backgrounds. (**C**) Histogram of the CheA-mTq foci distance to the nearest cell pole in wild-type and Δ*flhF* strain backgrounds. In (**B**) and (**C**) error bars depict counting error. (**D**) Percent of cells exhibiting polar localization of CheA-mTq in wild type and Δ*flhF* strain backgrounds from three biological replicates. Strains contain empty vector (pVC) or a complementing plasmid. Error bars depict the standard deviation. (**E**) Co-immunoprecipitation of CheA-mTq and Pch-VSV-G. CheA-mTq of the input lysates and anti VSV-G agarose bead elution fractions was detected by western blot utilizing a monoclonal anti-GFP antibody. Lanes 1–4 show control isolates lacking a CheA-mTq fusion. Lanes 1, 2 and 5, 6 show protein complexes eluted from anti VSV-G agarose beads.

*Figure 5. Continued on next page*

*Figure 5. Continued*

The following figure supplements are available for figure 5:

**Figure supplement 1**. CheA localizes to the same pole as FliM.

**Figure supplement 2**. CheA-mTq colocalizes with Pch-mCherry but not with FliM-mKate2.

**Figure supplement 3**. FliM-mKate2 polar localization is not dependent upon CheA.

effect of elevated levels of c-di-GMP on single cell motility has not been determined. It is known in *E. coli* and *S.* Typhimurium that c-di-GMP binds the YcgR protein and this complex interacts with the flagellar motor to reduce velocity and inhibit rotor reversal events (*Ryjenkov et al., 2006*; *Boehm et al., 2010*; *Paul et al., 2010*; *Zorraquino et al., 2013*). A *P. aeruginosa pch* mutant reduces swimming and swarming motility as determined by bulk, agar-based assays (*Li et al., 2007*; *Roy et al., 2012*). The end result of such assays can be affected by multiple indirect factors including production of rhamnolipids (*Caiazza et al., 2005*), genes that regulate Type IV pili (*Overhage et al., 2007*), chemotactic behavior (*Ferrandez et al., 2002*), and cell velocity (*Arora et al., 1996*). Therefore, it is not clear how c-di-GMP affects cell motility to alter the output of these bulk assays.

To further characterize the effect of c-di-GMP, we measured motility of single cells in a uniform environment. Exponential phase cells were prepared in the same manner as for c-di-GMP quantification. Cell motility was subsequently monitored in the same growth medium, but with the addition of viscous 1% methyl cellulose to reduce swimming velocity (*Schneider and Doetsch, 1974*) and facilitate the capture of cell trajectories by video microscopy. To exclude other forms of motility that require surfaces, we imaged cells at a focal plane 50 microns above the coverslip. We implemented an automated method of cell tracking and processing to quantify motility in an un-biased and large-scale fashion. We quantified mean cell velocity and determined that the majority of Δ*pch* cells were either amotile or traveled at a much lower velocity than wild type (*Figure 6A*). Conversely, ectopically reducing c-di-GMP to levels lower than wild type, utilizing a PDE expressing strain, increased the population of cells with mean velocities between four and eight microns/s and decreased the population of amotile cells. These results indicate that the c-di-GMP levels in wild-type cells have a physiological role in modulating cell velocity, as both reducing and increasing c-di-GMP levels affects velocity.

We also quantified reversals exhibited by cells in a uniform environment to determine the effect of c-di-GMP on chemotactic behavior. In this environment, cells utilize the chemotaxis machinery to exhibit reverse movement, not in response to a chemical gradient, but due to the effect of steady-state kinase activity. Approximately 25% of the wild-type cells exhibited at least one reversal in direction for trajectories of 25 microns (*Figure 6B*). Reversals were increased in cells ectopically expressing a PDE as at least 60% of cells exhibited one or more reversals. In contrast, the small percentage of motile Δ*pch* cells exhibited fewer reversals than wild type. Control strains, Δ*cheY* and Δ*cheB* gave the expected results in which more than 95% of Δ*cheY* cells exhibited no reversals and more Δ*cheB* cells relative to wild type exhibited increased reversals. These data indicate that c-di-GMP levels act to create heterogeneity not only in cell velocities but also in modulating chemotaxis machinery output. For organisms such as *E. coli*, the effect of c-di-GMP on reversals and velocity is mediated by the PilZ domain protein, YcgR that binds the flagellar rotor or stator upon forming a complex with cyclic di-GMP. We deleted the homolog of this gene in *P. aeruginosa*, PA3353, and found this gene did not have a significant effect on either velocity or reversal rate, as both were similar to that of wild type. This result suggests, in the conditions used for the assay, that this PilZ domain protein does not mediate reduction of cell velocity or inhibition of flagellar motor reversals by c-di-GMP.

## Discussion

The use of a biosensor that enables real time measurement of c-di-GMP in individual cells allowed us to define an unexpected mechanism by which the chemotaxis machinery controls c-di-GMP heterogeneity, thereby regulating flagellar-based motility. We identified the PDE activity of Pch as necessary for the maintenance of low c-di-GMP and for generating asymmetry in c-di-GMP levels following cell division. We also found that Pch is not sufficient to generate low c-di-GMP levels but additionally requires an intact flagellar apparatus as well as the chemotaxis machinery protein CheA for its localization

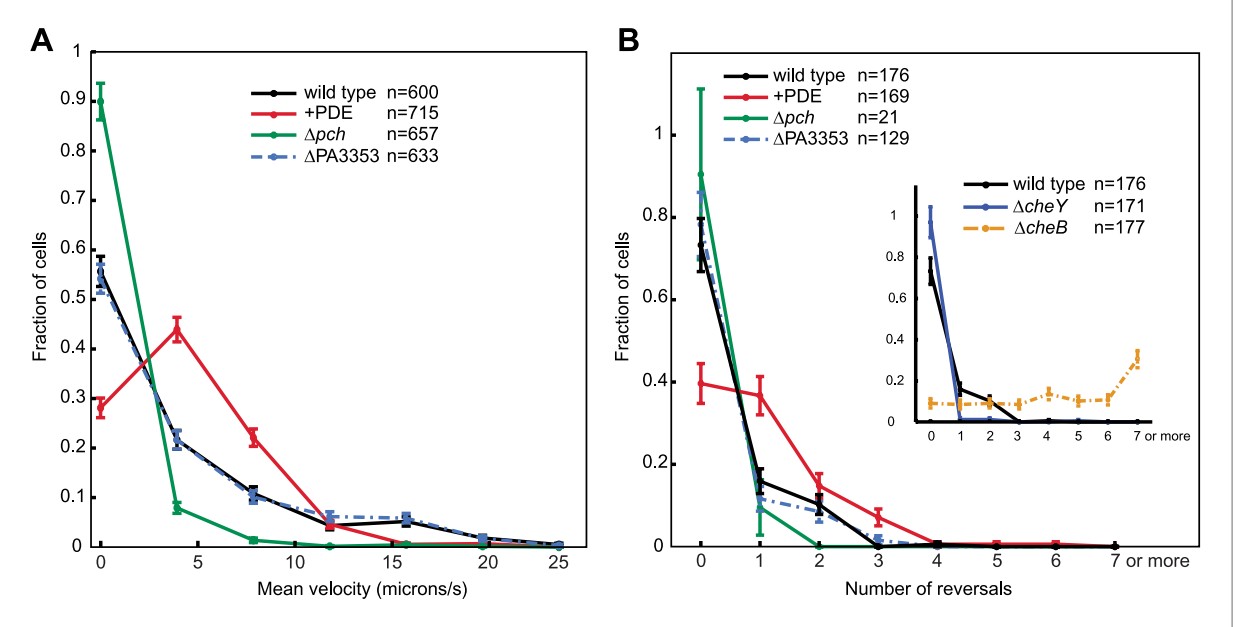

**Figure 6**. Cells with increased c-di-GMP exhibit decreased velocity and reversals. (**A**) Histogram of mean cellular velocity. The '+PDE' strain uniformly exhibits cellular c-di-GMP concentrations less than 65 nM. (**B**) Histogram of total number of reversals exhibited during 25 micron trajectories. Only cells moving at a velocity of 2 microns/s or greater were analyzed. Reversal data from the control strains are shown in the inset box. The *cheY* deletion is not expected to undergo any reversals. The *cheB* deletion is expected to undergo more reversals than wild type. Error bars depict the counting error. Representative graphs are shown for both (**A**) and (**B**).

and activity. Our results from imaging and co-immunoprecipitation studies suggest Pch binds to CheA or a CheA-associated protein, targeting it to the pole. As CheA is part of the chemotaxis machinery and is normally localized to one cell pole, this localizes Pch to the pole of a single daughter cell following cell division. Together these events contribute to differential c-di-GMP levels in daughter cells following cell division.

It is likely that complex formation between Pch and CheA occurs via an adaptor protein. Only CheW-like and response regulator receiver domains have been characterized to interact with CheA orthologs. Pch lacks such domains, and as receiver domains are typically phosphorylated by histidine kinases, it is unlikely Pch is phosphorylated by CheA. Pch possesses, in addition to an EAL domain that confers PDE activity, GAF, PAS, and PAC sensory input domains and a degenerate GGDEF domain. GGDEF domains confer DGC activity when functional, and have been shown to interact with receiver-like adaptor domains (*Chan et al., 2004*). This interaction suggests that an unidentified receiver domain protein could serve as an adaptor mediating an association between the Pch GGDEF domain and CheA. As receiver domains are often phosphorylated by histidine kinases, this putative adaptor could also modulate the control of Pch activity in response to CheA phosphorylation. One of the multiple domains in Pch may also modulate additional, uncharacterized protein–protein interactions. Evidence for additional interactions includes the observations that Pch-mCherry localization appears to be more diffuse in the *cheA* deletion than in the *fliF* deletion. FliF is required for assembly of a flagellar basal-body rod-hook structure that facilitates secretion of the FliA anti-sigma factor, FlgM (*Karlinsey et al., 2000*). FlgM secretion enables FliA mediated expression of the *P. aeruginosa* flagellar regulon Class IV genes (*Dasgupta et al., 2003*). Other, unidentified proteins within the FliA regulon, whose expression is abrogated in the *fliF* mutant, could therefore mediate the difference in Pch-mCherry subcellular localization observed between the *fliF* and *cheA* deletions. Such an interaction could facilitate the transport and localization of Pch to the chemotaxis machinery, or modulate its turnover in the cell. Interestingly, the Pch PDE has been shown to be activated by another molecule controlled by external signals, cAMP (*Roy et al., 2012*). We observed that a minority of cells exhibited high c-di-GMP concentrations, yet showed polarly localized Pch (*Figure 2B*). In these cells, Pch could be lacking a specific form of allosteric activation. Consistent with this possibility, Pch has several putative sensory input domains that could bind additional, yet to be identified allosteric effectors.

The requirement for the chemotaxis machinery to stimulate c-di-GMP turnover ensures that heterogeneity in c-di-GMP levels occurs preferentially in motile and chemotactic competent populations. c-di-GMP inhibits motility of *E. coli* and *S.* Typhimurium by reducing cell velocity and inhibiting tumbling events. In this study, we found that reducing cyclic di-GMP levels in the chemotaxis competent *P. aeruginosa* population enables increased cell velocities and reversals. Therefore, heterogeneity in c-di-GMP levels within a population confers diversity in cell reversal rates and in cell velocities. Populations of motile *E. coli* cells have been noted to exhibit a variety of chemotaxis responses to chemical gradients (*Koshland, 1980*; *Spudich and Koshland, 1976*). These authors and others have speculated that populations with diversity in such behavior may have an advantage when encountering a variety of chemical gradients (*Vladimirov and Sourjik, 2009*; *Sourjik and Wingreen, 2012*). Although our observations were performed in a uniform environment, it is likely that the diversity in c-di-GMP levels in *P. aeruginosa* also generates variation in the chemotactic response to a variety of chemical gradients and may therefore be a strategy to maximize survival in unpredictable environments.

This work indicates that population differences in the second messenger, c-di-GMP, are not simply stochastic but involve a specific molecular mechanism associated with the variation in cellular inheritance of specific organelles. In this regard, it is interesting to speculate that similar mechanisms for generating heterogeneity may be associated with other organelles such as pili, paralogous chemotaxis clusters, or flagella, which are inherited in an uneven fashion even in organisms with multiple flagellar organelles such as *E. coli* (*Ping, 2010*). Organelle-based partitioning of cellular proteins could therefore be a broader mechanism by which bacteria generate cellular diversity within populations after cell division.

## Materials and methods

### Bacterial isolates and plasmids

We used strain *P. aeruginosa* isolate PA14 to generate all results, except when mapped transposon mutants were utilized. Strain PAO1 was used when mapped transposon insertions were screened for effects on c-di-GMP and for effects on Pch-mCherry localization. In this instance, the Pch-mCherry fusion was introduced into the attTn7 site of PAO1. The 38 putative DGCs and PDEs of the *P. aeruginosa* strain PAO1 were identified by the presence of a GGDEF or EAL domain (*Jacobs et al., 2003*). The effects of abrogating flagellar components on Pch-mCherry localization were initially determined using the two-allele collection of mapped transposon insertions in PAO1 derived from *Jacobs et al (2003)* (*Held et al., 2012*). All figures utilize data generated from isolate PA14, with the exception of *Figure 3—figure supplement 1*. We found it necessary to use the PA14 isolate because *P. aeruginosa* PAO1 was susceptible to the loss of the biosensor encoding plasmid- and phage-induced lysis of colonies. All null mutations were created as in-frame deletion mutants in the PA14 isolate. The chromosomal in-frame deletions and point mutations were constructed at the native loci in the chromosome using allelic exchange with pEX18 Gm or pEX2Gm vectors (*Schweizer and Hoang, 1995*; *Rietsch et al., 2005*).

The CheA fusion to mTurquoise2 (mTq) (*Goedhart et al., 2012*) and the FliM fusion to mKate2 (Evrogen, Moscow, Russia) were constructed at the native chromosomal locations in PA14 using the pEX18 Gm system. These two fluorescent proteins were chosen for their better resistance to photobleaching, maturation, and brightness. The CheA-mTurquoise2 (mTq) C-terminal fusion was made by fusing the *cheA* gene with the fluorophore at the 3′ end using the GTGGSGGS encoding flexible linker. This fusion and roughly 1 kb of the upstream and downstream DNA sequence flanking the *cheA* gene were cloned into the pEX18 Gm vector. Identical strategies were used for creating the pEX18 Gm-derived N-terminal Flim-mKate2 fusion vector. The mKate2 gene was inserted at the 5′end of the *fliM* gene minus the start codon, using the EAAAK(x4) encoding linker with the same restriction sites. These pEX18 Gm vectors were subsequently used in allelic exchange procedure to construct fusions in the native chromosomal loci.

Plasmids to complement deletions of the *fliF*, *cheA*, and *flhF* genes in PA14 were created by designing primers to amplify those genes by PCR, including restriction sites at the ends of the primers for subsequent digestion and ligation into the pMMB67EH-GM vector (*Furste et al., 1986*). The PCR, digestion, and gel-purification were all performed using standard methods.

PA14 derivatives encoding an inducible ectopic DGC or PDE contain the *C. crescentus* DGC, CC3285, or PDE, CC3396, at the chromosomal Tn7 phage attachment site, attTn7, with the expression of cloned genes under the control of the pBAD promoter. The vector used to construct these strains

is based off of the pUC18Tmini-Tn7T vector (*Choi and Schweizer, 2006*), and was modified to contain the pBAD (arabinose regulated) promoter, the transcriptional regulator *araC*, and arabinose permease, *araE*. The genetic region containing *araC*, the pBAD promoter, the multiple cloning site and the T1T2R5 transcriptional terminator was PCR amplified from pBAD18. To facilitate transport of arabinose into *P. aeruginosa*, a DNA fragment containing the pCP13 promoter and the *araE* gene was amplified from pJAT13-*araE* (*Guzman et al., 1995*; *Khlebnikov et al., 2001*). These two fragments were ligated and cloned into Stu1-Spe1 sites in the pUC18TminiTn7TGm vector. *P. aeruginosa* neither metabolizes nor transports L-arabinose into the cell. Thus, the presence of the AraE transporter facilitates uniform transport of arabinose into every cell in the population, assuring gene induction in each cell (*Khlebnikov et al., 2001*).

The *in trans* chromosomal fusions of *pch* to genes encoding mCherry and SYFP2, referred to as Yfp (*Kremers et al., 2006*), are located at the chromosomal attTn7 site and were constructed using pUC18Tmini-Tn7T (*Choi and Schweizer, 2006*). The *pch* deletion was complemented by introducing this gene into the attTn7 site (using the pUC18Tmini-Tn7T system). All variants of *pch* were cloned to include its native promoter located in the 610 base pair segment upstream of the gene. A GGSGGS encoding linker was used in between *pch* and the gene encoding fluorescent protein. The E675A catalytic mutation was created by mutating codon 675 gAa to gCa in *pch* cloned into pUC18Tmini-Tn7T.

## Culture conditions

Antibiotic concentrations used for *P. aeruginosa* strain construction were 30 µg/ml gentamycin, 150 µg/ml carbenicillin, and 25 µg/ml Irgasan. Antibiotic concentrations used for maintenance of plasmids in *E. coli* were 15 µg/ml gentamycin and 100 µg/ml ampicillin. Both the bacterial species were routinely grown in LB medium.

## Bulk motility assay

Overnight cultures were diluted to an OD600 of 0.01. 1 µL of this suspension was spotted on motility medium consisting of 0.3% agar, 1% tryptone, and 0.5% NaCl and incubated at 30°C for ~14 hr.

## Image acquisition

Images were acquired using Nikon Elements AR software on a Nikon Ti-E (inverted) using a Nikon100X oil CFI Plan Apochromat λ DM objective (1.45NA) (Nikon Instruments, NY), Cascade II 1024 EMCCD camera (Photometrics, Tucson, AZ) and a 300W Xenon lamp (Lambda SL, Sutter instruments, Novato, CA) as the light source. mCyPet (Cfp), Fret, and mYPet (Yfp) images were collected using Semrock filters (Semrock Inc, Rochester, NY) housed in external filter wheels (Sutter Instruments), and dichroics as described in *Christen et al. (2010)* For snapshot FRET measurements, Cfp (mCyPet) and Fret channel images were collected using 600 ms exposure times with 2 × 2 binning and a multiplier of 3035. Yfp (mYPet) images were collected using 100 ms exposure with 2 × 2 binning and the same multiplier. For timelapse imaging, exposure times for Cfp and Fret images were reduced to 300 ms and the Yfp image was omitted to reduce photobleaching. mCherry and mKate2 fluorescence were collected using a filter cube designed for mCherry fluorescence (mCherry-40LP-A-000-ZERO, Semrock) with an exposure time of 1 s and 2 × 2 binning for mCherry (except during timelapse acquisition where the exposure time was reduced to 800 ms) and 1 × 1 binning for mKate2 (2 × 2 binning was used for *Figure 5—figure supplement 1B*). mTurquoise and SYFP2 were imaged using the filter set used for Cfp and Yfp, respectively. For both, 1 × 1 binning and exposure times of 600 ms and 1.2 s were used. Snapshot images were taken within 10–25 min of harvesting the cells from liquid culture. A minimum of four fields was acquired for each data set where at least three biological replicates were collected for each data set. All images were collected in a temperature controlled chamber set to 25°C. To facilitate segmentation and protein localization analyses, and to reduce possible effects on c-di-GMP from increased density, fields with a maximum of 200 cells were analyzed.

## Microscopy culture conditions

For microscopy, all strains containing the pMMB67 Gm vector control or with various inserts (biosensor or complementing clone) were grown overnight in the presence of 100 µM IPTG and 30 µg/ml gentamycin in LB medium. The next day, cultures were diluted to 0.05 in the same medium and grown in a baffled flask with shaking at 275 rpm at 30°C. For microscopy with the inducible PDE located at the attTn7 site under control of L-arabinose induction, L-arabinose was added to a final concentration of 0.2% to the overnight culture and at the time of sub-culturing. Cells were harvested at an OD600

of 0.3, pelleted, and resuspended in one fourth of the original volume in a medium formulated for microscopy because of its low intrinsic fluorescence consisting of 0.5 × M63, 10 mM succinate, 20 mM mono potassium glutamate, 1 × NEAA (50× and 100×) (Life Technologies, Grand Island, NY), 2 mM MgCl$_2$, and 100 μM Fe$^{2+}$NH$_4$SO$_4$ (*Christen et al., 2010*) 1.5 μL of this suspension was placed on a 0.75-mm thick 1% agarose pad containing the same microscopy medium and a coverslip was placed over the cell suspension before it was allowed to dry. The coverslip–agarose slide sandwiches were sealed using thermoplastic glue. Amine Labeling was performed as in *Skerker and Berg (2001)* using Alexa Fluor 488 Carboxylic Acid, Succinimidyl Ester (Life Technologies).

## Contrast parameters used for display of images

As we imaged protein–fluorophore fusions that are expressed under control of the native promoter and present as a single copy, it was necessary to adjust the brightness of the images to visualize fluorophore localization. Identical adjustments were made for images of the same protein–fluorophore fusions in all strain backgrounds. To adjust brightness in fluorescence images, we altered the parameters of a transfer function defined by the three parameters of an image's lookup table (LUT), the minimum, maximum, and gamma values. The maximum LUT value of these 16-bit images was chosen to be just greater than the 99th percentile of the pixel intensity distribution from the wild-type strain of a protein–fluorophore set. A protein–fluorophore fusion in the wild type background typically exhibited the greatest signal intensity. For Pch-mCherry, this resulted in a LUT range of 0–10,000. The timelapse image of Pch-mCherry was collected using an exposure time of 800 ms instead of 1 s and microscopy was performed near the end of the lamp's lifetime, meaning that the illumination was not comparable to that of the other images. Therefore, we set the LUT range from 0 to 4500, utilizing selection parameters mentioned above. For Pch-Yfp, the LUT range was 0–4000, and for CheA-mTurquoise, the LUT range was 0–12,000. For FliM-mKate2, the range was from 0 to 5500. All LUTs utilized a gamma value of 0.8.

We selected different contrast adjustments for the fluorescence image when part of the phase/fluorescence channel overlay; however the same contrast parameters were still applied for all strains expressing the same protein–fluorophore fusion. It was necessary to use different contrast adjustments for this type of overlay because only a single color was used to represent fluorescence intensity and it was therefore more difficult to discern pixels with lower intensity. This is especially true for the strain backgrounds that result in a diffuse localization pattern, because the protein is effectively distributed to a much larger volume, reducing signal. The maximum LUT value was chosen such that the localization pattern of the strain with the most diffuse protein–fluorophore could be visualized and was typically 60–70% of the maximum LUT value above. For Pch-mCherry, this corresponded to a range of 0–6000 for all strains. All LUTs utilized a gamma value of 0.8.

## Data analysis

Cell segmentation and fluorescence intensity analyses were performed using custom written Matlab software that was used previously for analysis of bacterial single cell microscopy data in *LeRoux et al. (2012)*. Prior to analysis in Matlab, images for the measurement of biosensor activity were preprocessed in ImageJ. To correct sub-pixel registration defects both Cfp and Yfp images were aligned to the Fret image using the Plugin TurboReg (*Thevenaz et al., 1998*) and background was subtracted using the rolling ball subtract background menu option, automated by a custom written Plugin. These were the only changes made to the raw fluorescence intensity images. In Matlab phase images were used for automated segmentation (*LeRoux et al., 2012*). The segmentation defined the boundaries of individual cells used in all analyses of fluorescence intensities. The cells not expressing the biosensor (less than 5% of cells) were excluded from the analysis by removing those cells with Yfp average arbitrary fluorescence units of less than 1500. In this study, net Fret (nFret) is defined as the average Fret intensity minus the average Cfp intensity multiplied by the spectral bleedthrough coefficient for Cfp into the Fret channel. This was calculated for these exposure conditions for the cells expressing mCyPet alone to be 0.46. The original nFret equation subtracts bleedthrough from Yfp into the Fret channel as well (*Xia and Liu, 2001*). Because imaging mYPet significantly increases photobleaching over time, our calculation of nFret does not include subtracting Yfp from the Fret signal. Polar intensity was defined as the area residing within either the end of the cell along the long axis within one fourth of the length of the cell. Polar localization is defined by a locus intensity score of greater than 14 where the locus must be within 7 pixels of the cell pole. Individual fluorescent foci were isolated using a watershed-based

segmentation algorithm and fit using a Gaussian point-spread function. To ensure that only bright, localized foci were used in analysis, a score was assigned to each focus proportional to the ratio of the peak intensity value to the standard deviation of the fit function.

## Criteria for the identification of putative DGCs and PDEs involved in c-di-GMP heterogeneity

Transposon insertion mutants were characterized by the difference in the percentage of cells containing less than 200 nM c-di-GMP as compared to the parental strain, wild type PAO1. 72% of wild type PAO1 cells exhibit c-di-GMP levels less than 200 nM. 80% of transposon insertions exhibited a difference relative to wild type of 10%. The remaining mutants, except for those in PA5017, exhibited a difference of less than 15% relative to wild type. The PA5017 transposon mutants exhibited a difference of greater than 60%. We chose to follow up on this result in the strain PA14, because we observed that PAO1 cells lose expression of the biosensor. PA14 expression is stable, and, although fewer PA14 cells exhibit low levels of c-di-GMP, abrogating PA5017 still has a dramatic effect on c-di-GMP levels.

### Biosensor calibration

The binding curve of the c-di-GMP biosensor in response to ligand was determined in *Christen et al. (2010)* and exhibits a Hill coefficient of 2 and a $K_d$ of 195 nM at 25°C. It was previously verified that FRET from purified biosensor as measured by microscopy gives a dose-response binding curve similar to that as measured by a spectrofluorometer. Levels of FRET as measured by microscopy corresponding to saturated or unsaturated biosensor were determined by measuring biosensor activity in cells expressing the ectopic *C. crescentus* DGC or PDE. Points in between were interpolated using the binding curve mentioned above. To minimize error, only points between 10% and 90% saturation are marked on the scale. For a more in depth discussion of the calibration of biosensors, refer to *Bermejo et al. (2011)*.

### Photobleaching correction

To correct for photobleaching that occurs during time-lapse imaging, the Cfp and Fret channel images were multiplied by a correction factor that was determined by dividing the average cellular fluorescence of cells in that channel by the average cellular fluorescence at time zero. Fluorophores mYPet and mCyPet exhibit different photobleaching curves, meaning that the amount of FRET influences the rate at which bleaching occurs. Therefore, we determined bleaching correction factors by dividing the populations of cells into those that exhibit high or low FRET and took the average of these two numbers to prevent any bias in apparent bleaching that would occur because of disparate number of cells exhibiting high and low FRET.

### Single cell motility image acquisition

Cells were grown to an OD600 of 0.3 in LB, resuspended in fresh LB, and diluted 1:1 in 2% methyl cellulose. The cell suspension was placed in a chamber the depth of one coverslip (no 1.5) and sealed with apiezon grease, Type M (Apiezon, Manchester, United Kingdom). Atmospheric air was intentionally trapped in the chamber to ensure the medium would continue to be oxygenated. The cells were observed using a mock darkfield set-up combining a Ph3 condenser annulus with a 20× objective. With these settings, one pixel equals 0.533 microns. To ensure movement observed was due to flagellar motility and not twitching motility, cells were imaged 50 microns from the coverslip. A total of 1000 frames were collected at a rate of 20 frames/s using an Andor iXon3 897 camera (Andor, Belfast, United Kingdom).

### Analysis of cell trajectories

Unless otherwise stated, analysis was performed in Matlab. Images were aligned using the multimodality non-rigid demon algorithm registration package described in *Wang et al. (2005)*. To subtract large background artifacts, an average of all images was subtracted from each image, and further background subtraction was performed by rolling ball subtraction in ImageJ. Cells were identified by thresholding on intensity. Trajectories were assembled using the Matlab implementation, developed by Daniel Blair and Eric Dufresne (http://physics.georgetown.edu/matlab/), of the IDL particle tracking algorithm. Trajectories were discarded if they contained less than 40 frames. To determine idealized cell trajectories from the noisy cell trajectory data, we implemented a Change Point Analysis on cell position following the techniques described in *Watkins and Yang (2005)* and *Kalafut and Visscher (2008)*. Reversals

were then determined from cells with mean velocity of greater than 2 microns/s by comparing the angles of sequential time ordered line segments. A difference between sequential angles of 150–210° was called a reversal. Histograms representative of multiple replicates performed are plotted in *Figure 6*.

## Co-immunoprecipitation of CheA-GFP and pch-VSV-G

Co-IP studies were conducted by using an epitope-tagged Pch protein to pull down CheA. The epitope tag, VSV-G, was made to the C-terminal of Pch by cloning a codon optimized, short DNA fragment containing GGT ACC GGG GGC AGC GGC GGC AGC GGA TCC TAC ACC GAC ATC GAA ATG AAC CGC CTG GGC AAG sequence, which encoded the GTGGSGGSGT unstructured linker and the YTDIEMNRLGK epitope tag. The DNA encoding the Pch::VSV-G fusion protein was cloned into pMMB67EHGm and conjugated into *P. aeruginosa* PA14 Δ*pch* or *P. aeruginosa* PA14 Δ*pch* containing CheA-SYFP2 fusion (CheA-YFP) at the native location in the chromosome. For preparing bacterial cell lysates, bacteria containing either pMMB67EHGm-*pch*::VSV-G or the vector alone were grown in identical conditions to those used for microscopy studies with the following modifications: overnight cultures were resuspended in 50 ml of fresh LB broth, and the fusion protein expression was induced by using IPTG at 250 μM at 0.1 OD600 nm. The cultures were further grown until OD600 nm reached 0.5. Bacteria were spun down and resuspended in 1.5 ml TBS lysis buffer (50 mM Tris, pH 7.4; 150 mM NaCl, 20 mM DTT, protease inhibitor cocktail (Roche Diagnostics, Basel, Switzerland), Dnase1 and lysozyme), sonicated, pelleted, and the cell lysate was collected for further studies. Protein levels of each lysate were normalized to have identical concentrations for the pull down assay. Monoclonal Anti VSV-G agarose beads (Sigma Aldrich, St. Louis, MO, Catalog # A1970) were prepared according to the manufacturer's instructions, and 20 μl of beads were incubated with 1 ml of cell lysates overnight at 4°C. The next day, the beads were spun down and washed in 10 ml of cold TBS buffer six times (containing protease inhibitors and 10 mM DTT). Protein complexes were eluted from the beads by heating with SDS-PAGE gel loading buffer. Expression of Pch::VSV-G and the CheA-YFP in appropriate strains were confirmed by performing immunoblotting using anti-VSV-G antibody (Rabbit polyclonal, Sigma Aldrich, # V4888) and anti-GFP antibody (Thermo Scientific, Waltham, MA, mouse monoclonal # GF28R) and visualized in a Licor infrared scanner using DyLight 680-anti mouse and DyLight 800-anti rabbit secondary antibodies (Thermo Scientific, # SA5-10170 and # SA5-10044). For visualizing CheA-SYFP2 fusion for *Figure 4*, ECL-based (GE Healthcare, Cleveland, OH) immunoblotting was performed using rabbit anti-GFP antibody at 1:1000 dilution (GenScript, Piscataway, NJ, # A011388-40) and donkey anti-rabbit HRP conjugate (GE Healthcare, # NA9340).

## Acknowledgements

We thank Dr Caroline Harwood and Dr Joseph Mougous for helpful discussions, Mauna Edrozo for technical help, the laboratory of Dr Theodorus Gadella for providing the mTurquoise and SYFP2 encoding plasmids, Kiara Held and Dr Colin Manoil for providing transposon insertions, Dr Nathan Kuwada for writing a description of the locus finding algorithm, and Gary Davis and Dr Dan Fong from Nikon Instruments for support with microscope hardware and software.

## Additional information

### Funding

| Funder | Grant reference number | Author |
| --- | --- | --- |
| National Institute of Allergy and Infectious Diseases | 5U54AI057141-09 | Bridget R Kulasekara, Cassandra Kamischke, Hemantha D Kulasekara, Samuel I Miller |
| National Science Foundation Graduate Research Fellowship | 2007047910 | Bridget R Kulasekara |
| National Institutes of Health | 1R21NS067579-01 | Hemantha D Kulasekara, Matthias Christen, Samuel I Miller |

The funders had no role in study design, data collection and interpretation, or the decision to submit the work for publication.

## Author contributions

BRK, HDK, Conception and design, Acquisition of data, Analysis and interpretation of data, Drafting or revising the article; CK, MC, SIM, Conception and design, Analysis and interpretation of data, Drafting or revising the article; PAW, Conception and design, Analysis and interpretation of data

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
