## [Decision Letter]

Thank you for sending your work entitled “c-di-GMP heterogeneity is generated by the chemotaxis machinery to regulate flagellar motility” for consideration at *eLife*. Your article has been favorably reviewed by a member of the Board of Reviewing Editors and two reviewers who discussed their comments before reaching the following decision.

All three reviewers are enthusiastic about the study by Kulasekara et al., which investigates the mechanism underlying c-di-GMP heterogeneity in *Pseudomonas aeruginosa*. While most of the data are complete and convincing, the manuscript would be improved if the authors addressed the following:

1) The manuscript would come full circle if a direct link between CheA and PA5017 were established. The current data demonstrate that CheA and PA5017 function together because a CheA mutant shows loss of heterogeneity and both proteins co-localize at the pole. However, the data do not show that this effect is direct. Could the authors provide evidence that CheA interacts with PA5017, for example by performing in vivo pull down experiments?

If the effect is direct, the authors need to discuss the mechanism. It is intriguing that the response regulator CheY is not required but the kinase activity of CheA is. Based on the discussion in the text, the authors propose that CheA modulates the PDE directly, by interaction with one of its several domains. This is surprising because PA5017 appears to lack a receiver domain. Has a CheA-type protein ever been shown to phosphorylate non response-regulator domains in a physiologically meaningful process? It is easier to postulate that an as yet unidentified response regulator protein mediates signal from CheA to PA5017. Such regulation is documented and there are many proteins containing both a receiver domain and a DGC/PDE domain (PleD for example). While the identification of this receiver is beyond the scope of the manuscript, this possible mechanism should be mentioned.

2) Figure 3: What are the relative levels of PA5017 in the strains carrying mutations in the different flagella and chemotaxis genes (i.e., can any of the differences in effects be explained by differences in the levels of the PDE)? Also, can the authors explain the different localization patterns seen for the different mutants?

3) The fluorescence localization data is only shown as overlays with the corresponding phase contrast images. This is not good practice because mounting overlay images requires contrast adjustment, potentially masking significant fluorescent signals. Unprocessed fluorescent images should also be shown in addition to the overlay images. In addition, given that the measured distance of the separation of the CheA-mTurquoise/FliM-mKate2 is within the diffraction limit, the CheA-mTurquoise/PA5017-YFP pair and CheA-mTurquoise/FliM-mKate2 pairs cannot be compared.

---

## [Author Response]

*1) The manuscript would come full circle if a direct link between CheA and PA5017 were established. The current data demonstrate that CheA and PA5017 function together because a CheA mutant shows loss of heterogeneity and both proteins co-localize at the pole. However, the data do not show that this effect is direct. Could the authors provide evidence that CheA interacts with PA5017, for example by performing in vivo pull down experiments*?

We successfully performed in vivo pull downs that verify CheA and PA5017(Pch) are part of the same protein complex (Figure 5) indicating CheA and PA5017 form a complex.

*If the effect is direct, the authors need to discuss the mechanism. It is intriguing that the response regulator CheY is not required but the kinase activity of CheA is. Based on the discussion in the text, the authors propose that CheA modulates the PDE directly, by interaction with one of its several domains. This is surprising because PA5017 appears to lack a receiver domain. Has a CheA-type protein ever been shown to phosphorylate non response-regulator domains in a physiologically meaningful process? It is easier to postulate that an as yet unidentified response regulator protein mediates signal from CheA to PA5017. Such regulation is documented and there are many proteins containing both a receiver domain and a DGC/PDE domain (PleD for example). While the identification of this receiver is beyond the scope of the manuscript, this possible mechanism should be mentioned*.

The reviewers have an excellent point and we have modified the Discussion to reflect the possibility that PA5017 (Pch) both associates with and is activated via an adaptor protein. The fact that we identified a protein complex does not preclude this possibility. We have incorporated the discussion of such an interaction in the discussion, citing the example of PleD.

*2)*
Figure 3*: What are the relative levels of PA5017 in the strains carrying mutations in the different flagella and chemotaxis genes (i.e. can any of the differences in effects be explained by differences in the levels of the PDE)? Also, can the authors explain the different localization patterns seen for the different mutants*?

To determine whether any effects were due to altered levels of the PDE, we quantified intensity from PA5017-mCherry (Pch-mCherry) expressed in the different mutant backgrounds (Figure 3—figure supplement 2). We found that the *fliF* mutant had reduced levels of Pch-mCherry (PA5017-mCherry) however the *cheA* deletion did not. We have addressed these findings in the text.

*3) The fluorescence localization data is only shown as overlays with the corresponding phase contrast images. This is not good practice because mounting overlay images requires contrast adjustment, potentially masking significant fluorescent signals. Unprocessed fluorescent images should also be shown in addition to the overlay images*.

We have added images of fluorescence intensity to accompany all overlays. However, we disagree that contrast adjustment is not good practice when studying low copy number proteins in bacteria; in fact we found contrast adjustment was necessary to visualize any signal. The fact that an image requires contrast adjustment means that the average fluorescent signal is significantly less than the maximum intensity value for the image’s bit depth. Furthermore, adjusting contrast does not mask significant fluorescent signals if the contrast adjustments are made to represent the range of intensities within an image. We carried out such efforts, as outlined in the Materials and methods section. Additionally, we would like to remind the reviewers that we imaged all protein-fluorophore fusions that are expressed under native control of the promoter and present in single copy, rather than protein-fluorophore fusions expressed off of a multi copy plasmid from an inducible promoter. We intentionally chose this methodology as an optimal means for visualizing protein localization, as it preserves the native stoichiometry between the protein-fluorophore fusion and any interacting partners. As expected, we have maintained the contrast adjustment across each set of individual protein-fluorophore fusions. Finally, these images were included to illustrate the differences in protein-fluorophore localization; however, we hope that the more extensive analysis entailing the automated quantitation of protein localization will also be persuasive in convincing the reviewers of the validity of our results.

*In addition, given that the measured distance of the separation of the CheA-mTurquoise/FliM-mKate2 is within the diffraction limit, the CheA-mTurquoise/PA5017-YFP pair and CheA-mTurquoise/FliM-mKate2 pairs cannot be compared*.

We agree that the two pairs should not be compared because we used different sets of filters, which could have different degrees of error. To remedy this problem, we have repeated measurements comparing the CheA-mTurquoise/FliM-mKate2 and CheA-mTurquoise/PA5017mCherry (Figure 5—figure supplement 2). Using these two pairs of fluorophores allows us to use the same pair of filters for both strains. Therefore, the error introduced by using the same pair of filters should be identical, and the calculated distance between fluorophore centroids can then be compared. In this instance, the diffraction limit does not apply because we are comparing the distance between centroids from two different fluorophores. A similar type of measurement has been made in Yamaichi et al. (A multidomain hub anchors the chromosome segregation and chemotactic machinery to the bacterial pole, *Genes Dev*. 2012 October 15; 26(20): 2348–2360. PMCID: PMC3475806).